# Searching for Convolutions and a More Ambitious NAS

## Abstract

An important goal of neural architecture search (NAS) is to automate-away the design of neural networks on new tasks in under-explored domains, thus helping to democratize machine learning. However, current NAS research largely focuses on search spaces consisting of existing operations—such as different types of convolution—that are already known to work well on well-studied problems—often in computer vision. Our work is motivated by the following question: can we enable users to build their own search spaces and discover the right neural operations given data from their specific domain? We make progress towards this broader vision for NAS by introducing a space of operations generalizing the convolution that enables search over a large family of parameterizable linear-time matrix-vector functions. Our flexible construction allows users to design their own search spaces adapted to the nature and shape of their data, to warm-start search methods using convolutions when they are known to perform well, or to discover new operations from scratch when they do not. We evaluate our approach on several novel search spaces over vision and text data, on all of which simple NAS search algorithms can find operations that perform better than baseline layers.

## 1 Introduction

Neural architecture search is often motivated by the AutoML vision of democratizing ML by reducing the need for expert deep net design, both on existing problems and in new domains. However, while NAS research has seen rapid growth with developments such as weight-sharing (Pham et al., 2018) and "NAS-benches" (Ying et al., 2019; Zela et al., 2020), most efforts focus on search spaces that glue together established primitives for well-studied tasks like vision and text (Liu et al., 2019; Li & Talwalkar, 2019; Xu et al., 2020; Li et al., 2020) or on deployment-time issues such as latency (Cai et al., 2020). Application studies have followed suit (Nekrasov et al., 2019; Wang et al., 2020).

In this work, we revisit a broader vision for NAS, proposing to move towards much more general search spaces while still exploiting successful components of leading network topologies and efficient NAS methods. We introduce search spaces built using the **Chrysalis**,[1] a rich family of parameterizable operations that we develop using a characterization of efficient matrix transforms by Dao et al. (2020) and which contain convolutions and many other simple linear operations. When combined with a backbone architecture, the Chrysalis induces general NAS search spaces for discovering the right operation for a given type of data. For example, when inducing a novel search space from the LeNet architecture (LeCun et al., 1999), we show that randomly initialized gradient-based NAS methods applied to CIFAR-10 discover operations in the Chrysalis that outperform convolutions—the "right" operation for vision—by 1% on both CIFAR-10 and CIFAR-100. Our contributions, summarized below, take critical steps towards a broader NAS that enables the discovery of good design patterns with limited human specification from data in under-explored domains:

- We define the broad NAS problem and discuss how it interacts with modern techniques such as continuous relaxation, weight-sharing, and bilevel optimization. This discussion sets up our new approach for search space design and our associated evaluations of whether leading NAS methods, applied to our proposed search spaces, can find good parameterizable operations.

- We introduce **Kaleidoscope-operations** (K-operations), parameterizable operations comprising the Chrysalis that generalize the convolution while preserving key desirable properties: short

---

[1]Following Dao et al. (2020), butterfly-based naming will be used throughout.

description length, linearity, and fast computation. Notably, K-operations can be combined with fixed architectures to induce rich search spaces in which architectural parameters are decoupled from model weights, the former to be searched via NAS methods.

- We evaluate the Chrysalis on text and image settings where convolutions are known to work well. For images, we construct the ButterfLeNet search space by combining K-operations with the well-known LeNet (LeCun et al., 1999). For text classification, we generalize the simple multi-width model of Kim (2014). On both we evaluate several applicable NAS methods and find that single-level supernet SGD is able to find operations that come close to or match the performance of convolutions when searched from-scratch, while also improving upon them when warm-started.

- We conclude by examining the generality of our approach on domains where convolutions are the "wrong" operation. We first consider permuted image data, where the "right" operation is permutation followed by convolution, and observe that NAS methods applied to the ButterfLeNet search space yield an architecture that outperforms all fixed operation baselines by 8%. Next we consider spherical MNIST data of Cohen et al. (2018), where the "right" operation is the spherical convolution from the same paper. We consider the K-operation search space that generalizes their network and again find that it outperforms convolutions by more than 20%. Our results highlight the capacity of K-operation-based search spaces, coupled with standard NAS methods, to broaden the scope of NAS to discovering neural primitives in new data domains.

## 1.1 RELATED WORK

AutoML is a well-studied area, with most work focusing on the fairly small search spaces of hyper-parameter optimization (Bergstra & Bengio, 2012; Li et al., 2018) or on NAS (Elsken et al., 2019). In the latter case, it has been observed that the search spaces are still "easy" in the sense that random architectures can do reasonably well (Elsken et al., 2019; Li & Talwalkar, 2019). More recently, Real et al. (2020) demonstrated the possibility of evolving all aspects of ML—not just the model but also the training algorithm—from scratch. We seek to establish a middle ground in which search spaces are large and domain-agnostic but still allow the encoding of desirable constraints and the application of well-tested learning algorithms such as stochastic gradient descent (SGD).

Our main contribution is a family of search spaces that build upon K-operations, which generalize parameterized convolutions (LeCun et al., 1999). Most NAS search spaces only allow a categorical choice between a few kinds of convolutions (Liu et al., 2019; Zela et al., 2020; Dong & Yang, 2020); even when drastically expanded to include many types of filter sizes and other hyperparameters (Mei et al., 2020), the operation itself is not generalized and so these search spaces may not be useful outside of domains where convolutions are applicable. Beyond NAS, recent work by Zhou et al. (2020) uses a meta-learning framing (Thrun & Pratt, 1998) to study how to learn more general types of symmetries—beyond simply translational—from multi-task data. This transfer-based setup allows a clear formalization of learning such equivariances, though unlike NAS, it is not applicable to single-task settings. In addition, their technical approach does not generalize two-dimensional convolutions due to computational intractability, while our K-operations are indeed able to do so.

The above works search over spaces of *parameterizable* operations by delineating a set of architectural or meta parameters to define the space over operations that are separate from model weights, which *parameterize* the operations found. In contrast, other efforts seek to simply outperform convolutions by directly training more expressive models. This includes several that use linear maps based on butterfly factors as drop-in replacements for linear or convolutional layers (Dao et al., 2019; 2020; Alizadeh vahid et al., 2020; Ailon et al., 2020). Very recently, Neyshabur (2020) showed that a sparsity-inducing optimization routine can train fully connected nets that match the performance of convolutional networks and in the process the weights learn local connectivity patterns. However, none of these papers return parameterizable operations from a formally defined search space.

## 2 STATISTICAL AND OPTIMIZATION OBJECTIVES OF NAS

In this section we set up the statistical and algorithmic objectives of neural architecture search. This is critical since we seek a definition of NAS that encompasses not only categorical decisions but also learning primitives such as convolutions. We ignore latency and transfer considerations and instead focus on the statistical problem of learning to parameterize a function $f_{\boldsymbol{w},a} : \mathcal{Z} \mapsto \mathbb{R}$ so as to minimize its expected value $\mathbb{E}_z f_{\boldsymbol{w},a}(z)$ w.r.t. some unknown distribution over the data-domain $\mathcal{Z}$. Here $a \in \mathcal{A}$ are *architectures* in some search space $\mathcal{A}$ and $\boldsymbol{w} \in \mathcal{W}$ are *model-weights* of sufficient

dimensionality to parameterize any architecture in $\mathcal{A}$. Classically, $a$ is chosen by a human expert to aid the data-driven selection of $\boldsymbol{w} \in \mathcal{W}$, i.e. to have favorable statistical and optimization properties to ensure that, given a finite data set $S \subset \mathcal{Z}$, some learning algorithm $\texttt{Alg} : 2^{\mathcal{Z}} \times \mathcal{A} \mapsto \mathcal{W}$ will return $\boldsymbol{w}_a = \texttt{Alg}(S, a)$ s.t. $\mathbb{E}_z f_{\boldsymbol{w}_a,a}(z)$ is small. $\texttt{Alg}$ is usually some iterative gradient-based approximation like SGD to the empirical risk over $S$.

**NAS Algorithms and Optimization Objectives:** In contrast to the standard learning setting, NAS aims to select both $a \in \mathcal{A}$ and $\boldsymbol{w} \in \mathcal{W}$ on the basis of training data, reducing the need for human intervention to constructing a search space and implementing a search algorithm, both of which should be as amenable as possible to general-purpose, domain-agnostic designs. Assuming a given training method $\texttt{Alg}$, this leads to search objectives of the form

$$\arg \min_{a \in \mathcal{A}} \quad \sum_{z \in V} f_{\boldsymbol{w}_a,a}(z) \qquad \text{s.t.} \qquad \boldsymbol{w}_a = \texttt{Alg}(T, a) \qquad (1)$$

where $T, V \subset S$ are disjoint training and validation sets. The main challenge with this search objective is intractability: (1) $\mathcal{A}$ usually contains combinatorially many architectures and (2) evaluating even one requires one run of $\texttt{Alg}$. Modern NAS avoids these problems using two techniques: *continuous relaxation* and *weight-sharing* (Pham et al., 2018; Liu et al., 2019) First, the often-discrete architecture search space $\mathcal{A}$ is relaxed into a convex space of architecture parameters $\Theta \supset \mathcal{A}$ such that any $\theta \in \Theta$ is associated with some architecture $a \in \mathcal{A}$ via some *discretization mapping* $\texttt{Map} : \Theta \mapsto \mathcal{A}$. For example, $\Theta$ could consist of linear combinations of operations in $\mathcal{A}$ and $\texttt{Map}$ could select the one with the largest coefficient. After continuous relaxation, we can run search by updating $\theta$ using gradients w.r.t. $f_{\boldsymbol{w}_\theta,\theta}$, replacing the approximation error incurred by truncating the discrete optimization of (1) with the one incurred by running the discretization $\texttt{Map}$ on the output of

$$\arg \min_{\theta \in \Theta} \quad \sum_{z \in V} f_{\boldsymbol{w}_\theta,\theta}(z) \qquad \text{s.t.} \qquad \boldsymbol{w}_\theta = \texttt{Alg}(T, \theta) \qquad (2)$$

This addresses the first intractability issue, but computing the architecture gradient w.r.t. $f_{\boldsymbol{w}_\theta,\theta}$ is still prohibitively expensive due to the need to differentiate through $\texttt{Alg}$. Weight-sharing resolves this via another approximation that simply maintains a fixed set of shared weights $\boldsymbol{w}$ throughout search and updates $\theta$ using the gradient of $f_{\boldsymbol{w},\theta}$, which does not depend on $\texttt{Alg}$. This exploits the fact that changing $\theta$ directly affects the objective $f_{\boldsymbol{w},\theta}$ and so having different weights for different architectures, i.e. not sharing them, is not necessary to distinguish their performance (Li et al., 2020). The weight-sharing approximation to (2) leads to alternating update methods (Pham et al., 2018; Liu et al., 2019) in which gradient updates to $\boldsymbol{w}$ using data from $T$ are alternated with gradient updates to $\theta$ using data from $V$; one can also define a single-level, empirical risk minimization (ERM) objective

$$\arg \min_{\theta \in \Theta} \min_{\boldsymbol{w} \in \mathcal{W}} \quad \sum_{z \in S} f_{\boldsymbol{w},\theta}(z) \qquad (3)$$

which can also be solved by alternating gradient updates, just using the same data, or by joint gradient updates. Despite eschewing the usual data-splitting of most AutoML algorithms, this objective has been found to do well in certain NAS settings (Li et al., 2020).

**The Goals of NAS:** The use of weight-sharing, continuous relaxation, and ERM blurs the line between NAS and regular model training, since architecture parameters are optimized in much the same way as model weights during search. The remaining differences are due to post-search discretization, in which an architecture $a = \texttt{Map}(\theta) \in \mathcal{A}$ is recovered from the output $\theta \in \Theta$ of (2) or (3), and post-discretization re-training, in which new weights $\boldsymbol{w} = \texttt{Alg}(S, a)$ are obtained with the discrete architecture. We will blur the line even further by considering search spaces that do not require post-search discretization and thus may not need post-discretization re-training either.

It is thus useful to formally state our main objective, in alignment with standard NAS (Liu et al., 2019; Ying et al., 2019), which is to use the given data $T$ to find a 'good' architecture $a \in \mathcal{A}$, i.e., one such that with suitable model weights $\boldsymbol{w}_a$ we obtain a function $f_{\boldsymbol{w}_a,a}$ with low test error. In practice, suitable model weights can be obtained either via: (a) *offline evaluation* in which we train model weights directly after discovering $a$, i.e., $\boldsymbol{w}_a = \texttt{Alg}(T, a)$; or (b) *supernet evaluation* in which we leverage the model weights learned jointly with architectural parameters, as in (2) or (3).

For most instances of NAS with weight-sharing, offline evaluation achieves better test performance than supernet evaluation; this likely results from a combination of (1) overfitting while training $\boldsymbol{w}$

in-conjunction with relaxed architecture parameters $\theta$, and (2) lossiness when discretizing to obtain a valid architecture $a = \text{Map}(\theta)$ post-search (Liu et al., 2019; Dong & Yang, 2020). However, our proposed search spaces do not require discretization, and we will see that supernet evaluation sometimes outperforms offline evaluation, which we view as a benefit since it removes the need for re-training. Notably, while single-task supernet evaluation can be viewed as similar in spirit to regular model training, as we use $T$ to jointly train a weight-architecture pair with low test error, we can isolate the quality of an architecture alone by performing supernet evaluation in a transfer learning setting, which we explore in Section 4.

## 3 GENERALIZING CONVOLUTIONS WITH PARAMETERIZABLE OPERATIONS

The use of parameterized convolution operations to extract features from images is a major architectural breakthrough that has spearheaded most recent progress in computer vision (LeCun et al., 1999). Convolutions have also found numerous application in other domains such as text (Kim, 2014; Zhang et al., 2015), but they may not be the appropriate parameterized operation for all domains, or the right kind of convolution may not always be evident or easy to construct (Cohen et al., 2018). Our main question is whether NAS can find a good parameterized neural primitive for a given data domain. To investigate, we start with a minimal requirement: that NAS can find an operation that matches the performance of convolutions given computer vision data. This leads to our main contribution: a *Chrysalis* of parameterizable K-operations that generalizes the convolution while preserving many of its desirable properties: small description length, linearity, and efficiency. Substituting K-operations for operations such as convolutions in backbone architectures yields search spaces in which the goal is to search the Chrysalis for a good K-operation to use on the input data.

**Desirable Properties of Parameterizable Neural Operations:** We first define what a parameterizable operation is; this is crucial for distinguishing our process of searching for such operations compared to directly training model parameters for them.

**Definition 3.1.** *A* **parameterizable operation** $\text{Op}(\cdot, \cdot)$ *is a function with two inputs, i.e., data and model parameters* $\boldsymbol{w}$, *and outputs a* **parameterized function** $\text{Op}(\cdot, \boldsymbol{w})$.

For example, a linear layer is a parameterizable operation that takes as input a matrix and outputs the corresponding linear map; Neyshabur (2020) trains such layers in an attempt to recover convolutional performance. Another example is the Kaleidoscope layer (K-layer) of Dao et al. (2020), which is similar except the matrix is constrained to be a stack of one or more structured maps. Notably, these and other works (Dao et al., 2019; Alizadeh vahid et al., 2020; Ailon et al., 2020) *fix* the parameterizable operation $\text{Op}$ and learn its parameterization $\boldsymbol{w}$; in contrast, we search for a good $\text{Op}$. In particular, we formulate our search for a good $\text{Op}$ via a search over architecture encodings $a$. As discussed in Section 2, we give evidence that the NAS approach of separating the search for $a$ from the learning of $\boldsymbol{w}$ yields good operations $\text{Op}_a$.

Our aim is to construct a search space that generalizes the convolution $\text{Conv}(\cdot, \cdot)$ while preserving its favorable, domain-*independent* properties; the goal will then be to use NAS to find operations within this search space that have good domain-*dependent* properties. We identify three domain-independent properties possessed by functions $\text{Conv}(\cdot, \boldsymbol{w})$ returned by the parameterizable convolution for any fixed input dimension $n \geqslant 1$ and any model weights $\boldsymbol{w} \in \mathbb{R}^{\mathcal{O}(n)}$:

1. Short description length: parameterized function $\text{Conv}(\cdot, \boldsymbol{w})$ can be represented with $\mathcal{O}(n)$ bits.
2. Linearity: $\exists~\boldsymbol{A}_{\boldsymbol{w},n}$ s.t. $\text{Conv}(\boldsymbol{x}, \boldsymbol{w}) = \boldsymbol{A}_{\boldsymbol{w},n}\boldsymbol{x}$ for all inputs $\boldsymbol{x} \in \mathbb{R}^n$ and $\|\boldsymbol{A}_{\boldsymbol{w},n}\|_F \propto \|\boldsymbol{w}\|_2$.
3. Fast computation: given arbitrary $\boldsymbol{x} \in \mathbb{R}^n$ we can compute $\text{Conv}(\boldsymbol{x}, \boldsymbol{w})$ in time $\mathcal{O}(n)$.

These properties have many learning-theoretic and computational advantages. In particular, models with short description length have better generalization guarantees (Shalev-Shwartz & Ben-David, 2014, Theorem 7.7); linearity is also advantageous statistically due to its simplicity and may interact better with optimization techniques such as batch-norm and weight-decay (Ioffe & Szegedy, 2015; Zhang et al., 2019). The last property has clear importance in practice and is intimately connected with the first, since any linear transform with description length $\tilde{\omega}(n)$ must take time $\tilde{\omega}(n)$.

**The Chrysalis of Kaleidoscope Operations:** To maintain these properties we turn to *Kaleidoscope matrices* (K-matrices) (Dao et al., 2020), which are products of butterfly matrices (Parker, 1995). Each butterfly matrix of dimension $n \times n$ is itself a product of $\log n$ sparse matrices with a special fixed sparsity pattern, which encodes the recursive divide-and-conquer algorithms such as the FFT. Each butterfly matrix has a total of $O(n \log n)$ parameters and $O(n \log n)$ runtime to

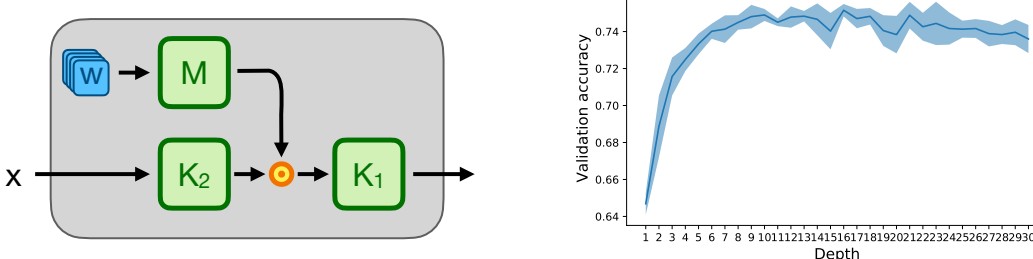

Figure 1: Illustration of a K-operation parameterized by weights $w$ and K-matrices $a = (K_1, K_2, M)$, with elementwise product in orange (left). $a$ is found via search, while weights $w$ are retrained for the discovered $a$. Plot of supernet evaluation of K-operations found when searching ButterfLeNet as a function of depth (right).

multiply a vector $x \in \mathbb{R}^n$. The *depth* of a K-matrix refers to the number of butterfly matrices in the product that form that K-matrix. Remarkably, all $n \times n$ matrices whose matrix-vector products can be expressed as a constant-depth arithmetic circuit with $\mathcal{O}(n)$ gates can be represented as a constant-depth K-matrix with memory and time complexity $\tilde{\mathcal{O}}(n)$ (Dao et al., 2020, Theorem 1); for many specific matrices, including convolutions, the logarithmic factors also disappear. These matrices thus provide a simple way to define a search space containing operations sharing the desirable properties of convolutions.

Several works have proposed a parameterizable operations based on butterfly factors; for example Dao et al. (2020) propose replace convolutional layers by K-matrices. As discussed above, such approaches *fix* a parameterizable operation and train model weights for it rather than finding one in a larger search space. In contrast we propose to search over a space of *kaleidoscope operations*, imposing the constraint that our selected K-operations have same *architecture* parameterizations across all instantiations in the resulting network, but can be parameterized by different model weights in each instantiation. This is analogous to the parameterization of the same convolution operation by different models weights (or filters) at each instantiation in a network. These K-operations comprise the Chrysalis, which can be combined with backbone architectures to create new search spaces containing potential operations for new domains.

For simplicity, we motivate and formalize our design of K-operations for the case of 1d input data; for higher dimensions, see Appendix A. To motivate our construction, note that a convolution of an input $x \in \mathbb{R}^n$ with filters $w \in \mathbb{R}^n$ can be expressed using the Fourier transform matrix $F \in \mathbb{C}^{n \times n}$:

$$\mathbf{Conv}(x, w) = F^{-1} \operatorname{diag}(Fw) Fx \qquad (4)$$

Thus we can obtain a space of parameterizable operations that contains convolutions by varying the matrices in the expression above. Since $F$ and its inverse are both K-matrices (Dao et al., 2020), this can be done efficiently as follows:

**Definition 3.2.** *A **Kaleidoscope operation** $\mathbf{K\text{-}Op}_a$ of depth $k \geqslant 1$ defined by three K-matrices $a = (K_1, K_2, M)$ of depth $k$ is an operation that, when parameterized by model weights $w \in \mathbb{R}^n$, takes as input an arbitrary vector $x \in \mathbb{R}^n$ and outputs $\operatorname{Real}(K_1 \operatorname{diag}(Mw) K_2 x)$.*

It is easy to show that K-operations satisfy the three desirable data-independent properties of convolutions. Furthermore, in the single-channel case the extended function class $\mathbf{K\text{-}Op}_a(\cdot, w)$ of depth $k$ can trivially express any depth-$2k$ K-matrix transform by setting $w = \mathbf{1}_n$ and $M = I_n$; this includes well-known functions such as Fourier, cosine, and wavelet transforms. Note that depth increases the expressivity of the space of K-operations and can be critical for performance (c.f. Figure 1). In the multi-channel case a layer of K-operations is less expressive than the K-layer of Dao et al. (2020) but can still express average-pooling (a type of convolution) and skip-connections, thus comprising most of operations in NAS search spaces such as DARTS (Liu et al., 2019).

The caveat here is that the Chrysalis itself does *not* contain these operations since it does not include any nonzero operations that ignore the filter weights $w$, which is required for parameter-free operations such as average pooling and skip-connections. This can be easily rectified by adding per-channel bias terms inside the $\operatorname{diag}$ that allow its output to be the identity, setting up a channel-connection problem related to that of Wortsman et al. (2019). However, we did not find simple algorithms for doing so empirically beneficial and so leave further exploration to future work.

Table 1: Comparison of architectures found by searching ButterfLeNet on CIFAR-10. Results are averages over five random seeds affecting both search and offline evaluation. When warm starting with convolutions, offline evaluation of Supernet SGD outperforms fixed convolutions by $1\%$ on both CIFAR-10 and on transfer to CIFAR-100. Furthermore, when search is initialized to a random operation, offline evaluation of Supernet SGDR matches the performance of fixed convolutions. Finally, we find that when warm starting with convolutions, supernet evaluation of ButterfLeNet using Supernet SGDR attains the best performance.

| CIFAR | fixed operation baselines | | | | DARTS | | Supernet SGD | | Supernet SGDR | |
|---|---|---|---|---|---|---|---|---|---|---|
| # classes (eval.) | | | | random | 1st | 2nd | from | warm | from | warm |
| | linear | conv | K-layer | K-op | order | order | scratch | start | scratch | start |
| 10 (supernet) | - | - | - | - | 51.25 | 70.25 | 75.27 | 76.43 | 76.84 | **77.15** |
| 10 (offline) | 59.17 | 75.76 | 68.64 | 56.76 | 57.92 | 71.88 | 73.75 | **76.46** | 75.43 | 75.98 |
| 100 (offline*) | 28.02 | 43.88 | 39.26 | 29.47 | 29.50 | 41.70 | 42.86 | **44.86** | 44.33 | 44.20 |

* For DARTS, Supernet SGD, and Supernet SGDR, a K-Op found on CIFAR-10 is transferred to CIFAR-100.

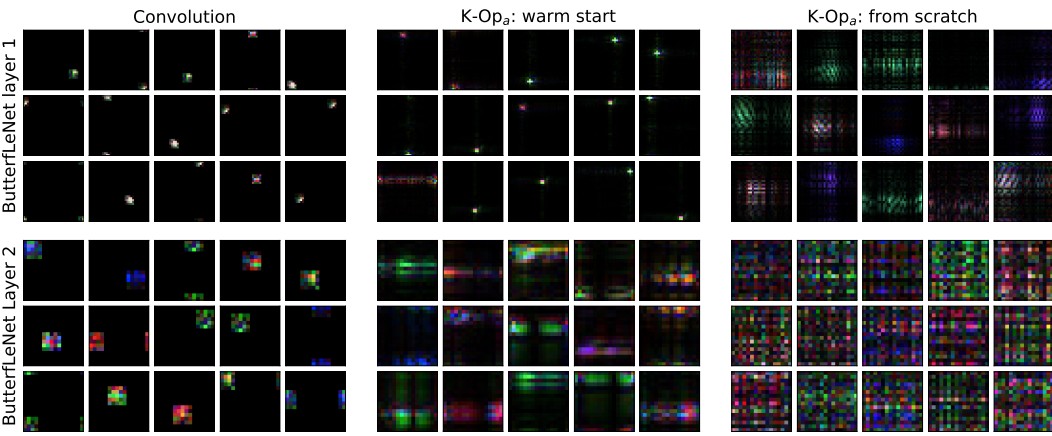

Figure 2: Comparison of the filters, and filter analogues, produced by three parameterized operations trained on CIFAR-10: convolutions, K-operations warm-started with convolutions, and K-operations searched from scratch. The latter two are discovered using Supernet SGD. In the case of convolutions, these are randomly selected filters at random positions on the input feature map. In the cases of the learned K-operations, these are randomly selected neighborhoods of the operation over the input feature map. The first layer of the warm started K-operation remains mostly convolutional, while the second layer pools features more globally. The first layer of the from scratch K-operation has some locality structure, while the second layer does not.

# 4 USING THE CHRYSALIS TO COMPETE WITH CONVOLUTIONS

We now examine the viability of the Chrysalis to find good K-operations on tasks where convolutions are known to perform well. We first consider standard computer vision tasks, i.e., the canonical application of convolutions, and next study another well-known application of parameterized convolutions: extracting features from temporal data such as text (Kim, 2014; Zhang et al., 2015).

**ButterfLeNet-Searching for Good Operations on Image Data.** To construct a search space from (2d) K-operations, we use the classic LeNet architecture (LeCun et al., 1999), which consists of two convolutional layers followed by two fully connected layers. We replace each convolutional layer by a layer in which the same depth-9 **K-Op** is shared across all input-output channels; our task on the resulting search space, **ButterfLeNet**, is thus to learn two operations in total, one for each layer. Offline evaluation of the discovered operation is conducted by fixing the architecture parameters $a$ and retraining the model weights using SGD with the same hyperparameters used for LeNet.

Because K-operations comprise a continuous search space, ButterfLeNet is amenable to standard gradient based optimization without the need for continuous relaxation. At the same time, it is not straightforward to define sampling-based approaches beyond random search, which itself works poorly due to the difficulty of the search space. We thus evaluate the following gradient-based methods, each with the same fixed budget in terms of number of training epochs:

- DARTS (Liu et al., 2019): approximates the bilevel objective (2) using Adam (Kingma & Ba, 2015) to update the architecture parameters and SGD to update the shared weights.

Table 2: Comparison of fixed operations and K-operations found via Supernet SGD over ButterfLeNet-Text. Tasks comprise a standard set of text classification benchmarks. Either supernet or offline evaluation of warm started K-operations found using Supernet SGD outperform the fixed operation baselines on all 7 datasets.

|  | method | evaluation | CR | MPQA | MR | SST1 | SST2 | SUBJ | TREC |
|---|---|---|---|---|---|---|---|---|---|
| baseline | **linear** | offline | 73.89 | 88.37 | 71.22 | 38.93 | 77.66 | 88.08 | 86.76 |
| operations | **convolution** | offline | 81.47 | 90.53 | 87.75 | 45.03 | 85.35 | 91.66 | 92.48 |
| | from scratch | supernet | 78.74 | 90.51 | 77.43 | 43.71 | 82.06 | 90.78 | 89.44 |
| **Supernet** | from scratch | offline | 78.58 | **91.24** | 76.09 | 43.74 | 82.56 | 90.40 | 90.44 |
| **SGD** | warm start | supernet | 83.11 | 90.84 | **80.00** | 44.91 | **85.84** | 91.88 | **93.20** |
| | warm start | offline | **83.74** | 90.90 | 79.72 | **45.24** | 85.3 | **92.18** | 92.24 |

- Supernet SGD: direct optimization of the single-level objective (3) using SGD.

- Supernet SGDR: runs Supernet SGD but periodically resets the step-size schedule and reinitializes model weights $w$; inspired by the warm-restart method of Loshchilov & Hutter (2017), it attempts to boost from-scratch performance by "warm-starting" from a discovered K-operation.

We further compare our discovered K-operations to four natural parameterizable operations: linear (fully-connected) layers, convolutions, K-layers, and random K-operations.

In Tables 1 we show that Supernet SGD is able to search ButterfLeNet from random initialization to find K-operations that match the performance of convolutions, i.e. regular LeNet. Despite being less expressive, our approach also outperforms the fixed K-layer approach in which all convolutions are replaced with fully-parameterized K-matrices. Interestingly, supernet evaluation is worse for established NAS methods like DARTS but better for direct optimization, with Supernet SGD-R outperforming convolutions by around 1.1% from scratch and 1.4% when warm-started. Because of the K-operation construction, we can also investigate the quality of K-operations found when we "warm-start" $\text{K-Op}_a$ to be a convolution by setting the K-matrices comprising $a$ to the appropriate Fourier transforms; doing so we find K-operations that outperform convolutions. Table 1 also shows a transfer learning experiment that further demonstrates the usefulness of the K-operations discovered on CIFAR-10: when the architecture parameters are fixed and the weights retrained on CIFAR-100, our operations outperform convolutions at test-time. Finally, we note the difficulty of our search space, as demonstrated by the poor performance of random K-operations, which do worse than even linear layers; this is in contrast to most NAS search spaces, which are often easy-enough to be solved by random search (Li & Talwalkar, 2019; Yang et al., 2020). We find that a larger ButterfLeNet model, Wide ButterfLeNet, can still outperform convolutions in terms of supernet evaluation. This is shown in Appendix B.

We also explore what operations are being learned, comparing convolutions, K-operations discovered by warm-starting with convolutions, and K-operations found from-scratch in Figure 2. Our visualizations suggest that learned K-operations use more global information to extract features, especially in the second ButterfLeNet layer discovered from-scratch.

**ButterfLeNet-Text: Searching for Good One-Dimensional Operations.** We next consider applications of temporal data such as text. We again use the Chrysalis to replace convolutions in all input-output channels in an existing model, namely the network of Kim (2014) which has three parallel convolutional layers with different kernel widths followed by a fully-connected layer; our task will be to find a separate K-operation for each of layer.

Our evaluation focuses on a standard suite of sentence classification tasks, with results presented in Table 2. As before, we compare the performance of Supernet SGD with the baseline performance of fixed linear operations. In 5 out of 7 of the datasets considered, offline evaluation of Supernet SGD where the K-operations are warm started yields higher performance than all fixed operation baselines. On the remaining two datasets, supernet evaluation outperforms convolutions and offline evaluation matches the performance of convolutions. We find that when the K-operations are initialized from scratch, offline evaluation substantially outperforms the fully connected baseline and, among the architectures considered, achieves the highest test accuracy on the MPQA dataset.

Table 3: Comparison of fixed operation baselines to ButterfLeNet trained using Supernet SGD on permuted CIFAR-10 and CIFAR-100. K-operations trained from scratch outperform all other methods on supernet evaluation, offline evaluation, and transfer to CIFAR-100.

| dataset | fixed operation baselines | | | Supernet SGD | |
|---|---|---|---|---|---|
| | linear | conv | K-layer | from scratch | warm start |
| CIFAR-10 (supernet) | - | - | - | **74.69** | 70.22 |
| CIFAR-10 (offline) | 59.61 | 58.90 | 66.16 | **72.99** | 69.56 |
| CIFAR-100 (offline∗) | 27.89 | 31.41 | 37.36 | **42.73** | 40.42 |

\* For Supernet SGD, a K-Op found on CIFAR-10 is transferred to CIFAR-100.

Table 4: Comparison of spherical convolutions, convolutions, and NAS on the ButterfLeNet-Spherical search space, built atop the convolutional baseline of Cohen et al. (2018). We test on the stereographically projected spherical MNIST as well as a rotated variant (Cohen et al., 2018).

| Spherical MNIST | fixed operations | | Supernet SGD | | |
|---|---|---|---|---|---|
| subset (evaluation) | conv | spherical | from scratch | warm start | warm start, from scratch |
| non-rotated (supernet) | - | - | 98.44 | 97.26 | 98.56 |
| non-rotated (offline) | 97.59 | 96.49 | 98.55 | 98.23 | **98.87** |
| rotated (supernet) | 33.92 | **96.32** | 35.77 | 28.26 | 56.11 |
| rotated (offline) | 33.92 | **96.32** | 33.38 | 30.29 | 54.49 |

# 5 BEYOND CONVOLUTIONS AND TOWARDS A MORE AMBITIOUS NAS

Convolutions are well-suited to many applications, but they are not always the best operation to use. Since our broader goal is to enable users to apply the Chrysalis to find neural primitives for whatever type of data they use, we now ask whether our novel search spaces can be leveraged to find operations that outperform convolutions in domains where they are *not* optimal. We establish this in two simple settings using the same approach as before: borrowing existing network structures and replacing convolutions by K-operations.

**ButterfLeNet: Unpermuting Image Data**. Here we consider a setting similar to that of Section **??** except a fixed permutation of all rows and columns is applied to CIFAR images before being passed as input. Since K-matrices can express both convolutions and permutations, there exist K-operations that do well on this data; this experiment thus tests whether we can leverage our search space to identify these good operations. Note that Dao et al. (2019) report a related experiment, in which they directly attempt to recover a permutation; our setting is more difficult because we are simultaneously attempting to do well on a classification task. Table 3 shows that both supernet and offline evaluation of Supernet SGD can attain nearly the same performance as in the unpermuted case when searching from-scratch. Perhaps unsurprisingly, warm-starting from convolutions performs worse. We also outperform all fixed linear operations, including the K-layer approach that is more expressive than our approach but experiences a slightly *larger* drop in performance from Table 1.

**ButterfLeNet-Spherical**. Finally, we consider the spherical MNIST dataset of Cohen et al. (2018), applying the Chrysalis to their baseline model consisting of two convolutional and one fully-connected layer. The spherical MNIST dataset consists of a stereographic projection of the MNIST dataset onto the sphere and a projection back to the plane, resulting in a nonlinear distortion. The rotated variant additionally applies random rotations before projecting back to the plane. We compare convolutions to search spaces induced by the Chrysalis. Since the spherical CNN of Cohen et al. (2018) uses two types of spherical convolution layers, we add an additional evaluation that only warm starts the first layer to break the symmetry of initializing both operations to be the same.

Table 4 shows that supernet and offline evaluation of Supernet SGD both find operations that significantly outperform convolutions but are also significantly behind spherical convolutions on the rotated spherical MNIST dataset. While convolutions, spherical convolutions, and K-operations all achieve high performance on the stereographically projected data, spherical convolutions are specifically designed to be invariant to rotation as well. It is unclear to what degree K-matrices are expressive enough to capture rotational invariance of stereographic projections. On the other-hand, our general-purpose approach significantly exceeds simple convolutions. This demonstrates that while it is difficult to match sophisticated operation designs, K-operations can still lead to strong improvements over convolutions in new domains.

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

Table 5: Performance on CIFAR-10 of a larger variant of LeNet ("Wide LeNet") with a greater number of channels. Here, supernet evaluation of warm started Supernet SGD achieves the highest performance on CIFAR-10.

| | | Supernet SGD | |
| dataset | conv | from scratch | warm start |
| --- | --- | --- | --- |
| CIFAR-10 (supernet) | - | 78.42 | **87.27** |
| CIFAR-10 (offline) | 86.72 | 76.51 | 86.72 |

## A HIGHER-DIMENSIONAL KALEIDOSCOPE OPERATIONS

To generalized Kaleidoscope operations to higher dimension, we simply perform the Kaleidoscope operations on each dimension of the input. This is similar to how 2d-FFT is done by performing two 1d-FFTs on each dimension of the input.

More precisely, we take the Kronecker product (denoted by $\otimes$) of the 1d Kaleidoscope matrices in the 1d Kaleidoscope operation. For simplicity of notation, we assume that the two dimensions have the same size $n$.

**Definition A.1.** *A* **2d Kaleidoscope operation** $\mathbf{K}\text{-}\mathbf{Op}_a$ *of depth* $k \geqslant 1$ *defined by six K-matrices* $a = (\boldsymbol{K}_{1,1}, \boldsymbol{K}_{1,2}, \boldsymbol{K}_{2,1}, \boldsymbol{K}_{2,2}, \boldsymbol{M}_1, \boldsymbol{M}_2)$ *of depth* $k$ *is an operation that, when parameterized by model weights* $\boldsymbol{w} \in \mathbb{R}^{n^2}$, *takes as input an arbitrary vector* $\boldsymbol{x} \in \mathbb{R}^{n^2}$ *and outputs* $\mathrm{Real}(\boldsymbol{K}_1 \operatorname{diag}(\boldsymbol{M}\boldsymbol{w})\boldsymbol{K}_2\boldsymbol{x})$ *where* $\boldsymbol{K}_1 = \boldsymbol{K}_{1,1} \otimes \boldsymbol{K}_{1,2}$, $\boldsymbol{K}_2 = \boldsymbol{K}_{2,1} \otimes \boldsymbol{K}_{2,2}$, *and* $\boldsymbol{M} = \boldsymbol{M}_1 \otimes \boldsymbol{M}_2$.

It is also straightforward to generalize to N-d:

**Definition A.2.** *An* **N-d Kaleidoscope operation** $\mathbf{K}\text{-}\mathbf{Op}_a$ *of depth* $k \geqslant 1$ *defined by* $3N$ *K-matrices* $a = (\boldsymbol{K}_{1,1}, ..., \boldsymbol{K}_{1,N}, \boldsymbol{K}_{2,1}, ..., \boldsymbol{K}_{2,N}, \boldsymbol{M}_1, ..., \boldsymbol{M}_N)$ *of depth* $k \geqslant 1$ *is an operation that, when parameterized by model weights* $\boldsymbol{w} \in \mathbb{R}^{n^N}$, *takes as input an arbitrary vector* $\boldsymbol{x} \in \mathbb{R}^{n^N}$ *and outputs* $\mathrm{Real}(\boldsymbol{K}_1 \operatorname{diag}(\boldsymbol{M}\boldsymbol{w})\boldsymbol{K}_2\boldsymbol{x})$ *where* $\boldsymbol{K}_1 = \otimes_{i=1}^N \boldsymbol{K}_{1,i}$, $\boldsymbol{K}_2 = \otimes_{i=1}^N \boldsymbol{K}_{2,i}$, *and* $\boldsymbol{M} = \otimes_{i=1}^N \boldsymbol{M}_i$.

## B WIDE BUTTERFLENET

We show that warm started supernet training of a larger LeNet architecture can outperform its fixed convolutional counterpart. These results are presented in Table 5.

## C EXPERIMENTAL DETAILS

### C.1 BUTTERFLENET-VISION

#### C.1.1 ARCHITECTURE DETAILS

For all LeNet experiments, we use the LeNet architecture except padding is added to the convolutional layers to preserve feature map dimension, and ReLU activations are used. Namely, this is convolution (3, 6), average pooling (2, 2), convolution (6, 16), average pooling (2, 2), linear (120), linear (84), linear (10 or 100). $5 \times 5$ kernels are used throughout. The linear baseline consists of 4 linear layers with ReLU activations. In particular, that is linear (6 * 16 * 16), linear (16 * 8 * 8), linear (120), linear (84), linear (10 or 100). The K-layer baseline replaces all convolutions in LeNet with K-layers of the same shape. A K-layer consists of two K-matrices per input-output channel pair. For ButterfLeNet experiments, we replace each convolutional layer in LeNet with a K-operation of the same shape, and parameterized by the same number of model parameters as the analogous convolutional layer. In particular for our warm start experiments, we initialize $K_1$ and $K_2$ to be inverse Fourier transform and Fourier transform matrices scaled to be unitary, while $M$ is initialized to be a unscaled Fourier transform matrix. Each of these are depth 1 K-matrices. When initializing K-operations from scratch, we use depth 9 K-matrices with unitary initialization. That is, $K_1, K_2, M$ are each a product of 9 K-matrices, where $K_1, K_2, M$ are each initialized to be random unitary matrices.

Wide LeNet comprises 3 convolutional layers with more channels than LeNet, and two linear layers. In particular, convolution (3, 32), batch norm, convolution (32, 64), batch norm, convolution (64,

128), linear (128), batch norm, linear (10 or 100). Here, convolutions use $3 \times 3$ filters. As before, ReLU activations are used. Wide ButterfLeNet replaces all convolutional layers with K-operations of corresponding dimension. In both the warm start and from scratch settings, we tie the operations between the second and third K-operation layers.

### C.1.2 Hyperparameters for offline evaluation and fixed baselines

For offline evaluation (except DARTS) and models with fixed operations, we train for 200 epochs using SGD with a learning rate of $0.01$, decreased to $0.005$ at epoch 100, to $0.001$ at epoch 150 and a minibatch size of 128. We use a weight decay of $0.0001$. For all methods except from scratch ButterfLeNet, we employ data augmentation with random cropping and random horizontal flipping. For from scratch ButterfLeNet, we instead employ the Fast AutoAugment policy found on CIFAR-10 (Lim et al., 2019). For offline evaluation of DARTS, we train using the Adam optimizer (Kingma & Ba, 2015) to stay faithful to the original DARTS formulation.

### C.1.3 Architecture search

For Supernet SGD methods, we train the supernet for 800 epochs using SGD and an initial learning rate of $0.01$, decreased to $0.005$ at epoch 400, to $0.001$ at epoch 600. We use the same augmentation scheme used for baselines and offline evaluation, where from scratch ButterfLeNet uses Fast AutoAugment. We use a weight decay of $0.0001$. First and second order DARTS use Adam (with a learning rate of $0.001$) for optimizing architecture parameters as well as a bilevel training routine. First order DARTS alternates between updating architecture parameters and model parameters using a validation set and the training set. In practice, we partition the training set into equally sized subsets. Second order DARTS uses a second order gradient with a 'lookahead' step, which is approximated using a finite difference approximation. We find that the second order update is more stable in our search spaces.

For Supernet SGDR methods, we train the supernet 4 times for 200 epochs each, using SGD with a learning rate of $0.01$, decreased to $0.005$ at epoch 100, to $0.001$ at epoch 150. All other hyperparameters are the same as above.

### C.2 ButterfLeNet-text

### C.2.1 Architecture details

Our convolutional baseline is the convolutional architecture proposed by Kim (2014) with static word2vec embeddings. This architecture involves three 1d convolutional layers in parallel with different filter sizes (3, 4, and 5), and 100 output channels each, which are concatenated. This is followed by 1d max pooling, and a $300 \times k$ linear layer where $k$ is the number of classes in the classification task. The linear baseline replaces the three parallel convolutional layers with a single linear layer. For both of these, the ReLU activation function is used throughout. ButterfLeNet-text, which generalizes the convolutional baseline, replaces all 1d convolutions with 1d K-operations of the same dimensionality. For from scratch, we use depth 9 K-operations.

### C.2.2 Hyperparameters for ButterfLeNet-text

We use the same training procedure for supernet training, offline evaluation, and fixed operations. In particular, we use the Adadelta optimizer (Zeiler, 2012) and train using early stopping with a patience parameter of 3 based on a held out validation set. We use a batch size of 50 and employ dropout with a probability of $0.5$ on the final linear layer.

### C.3 ButterfLeNet-spherical

### C.3.1 Architecture details

The baseline convolutional architecture that we consider comprises two convolutional layers and a fully connected layer. Both convolutional layers have $5 \times 5$ kernels with a stride of 3, the first has 20 output channels and the second has 40 output channels, and use ReLU throughout. This is followed

by a linear layer. ButterfLeNet-spherical replaces these convolutions with K-operations. We warm start the first K-operation as convolutions and initialize the second one from scratch.

### C.3.2 HYPERPARAMETERS FOR BUTTERFLENET-SPHERICAL

We use the same training procedure for supernet training, offline evaluation, and fixed operations. Namely, we train for 20 epochs using Adam with a learning rate of 0.0005 and a minibatch size of 32.

