# OpenReview forum: "Searching for Convolutions and a More Ambitious NAS"
_ICLR.cc/2021/Conference — Reject_

### Official Review · AnonReviewer4 · 2020-10-21
**The experiments and writing should be further improved**

**Rating:** 4
**Confidence:** 4

**Review:**

**Summary:**
The authors propose a new NAS method that comes with a space of operations generalizing the convolution. The proposed method makes it possible for users to design their own search spaces according to the training data. However, the experiments and writing should be further improved.

**Strengths:**
1. The proposed method enables users to build their own search spaces.
2. The authors incorporate Kaleidoscope-operations to develop a new search space.

**Weaknesses:**
1. The authors only conduct experiments on small datasets, e.g., CIFAR. It would be stronger to report the comparison results on ImageNet.
2. The authors seek to allow users to design their own search spaces adapted to their data, which, however, is very labor-intensive. Moreover, it is also not clear how different search spaces would affect the search performance?
3. The proposed method fixes the backbone architecture and only searches for the optimal operation. However, this method ignores the topology of architectures. In my opinion, architecture topology should be an important factor to search in NAS.
4. The authors argue that the proposed method “enables the discovery of good design patterns with limited human specification from data in under-explored domains”. What does “under-explored domain” mean? How to define it?
5. What is FFT in Page 4? There is no definition before it appears, which makes the paper hard to follow.
6. From Definition 3.2, the search process w.r.t. k-operations seems to learn the parameters of the matrix K1 and K2. Thus, it seems not a typical NAS process. It would be better to illustrate the relationship between the proposed method and the standard NAS.
7. It is not clear what the proposed search space is. How large is it?
8. Several state-of-the-art NAS methods should be compared in the experiments, such as PDARTS, PCDARTS, etc.
9. The writing of this paper should be improved. The paper is hard to follow and there are many grammatical errors:
(1) In Page 3, “the main challenge … is intractability” should be “The main challenge … is intractable”.
(2) In Page 5, “Dao et al. (2020) propose replace convolutional layers” should be “Dao et al. (2020) propose to replace convolutional layers”

---

> ### Author Response · Authors · 2020-11-20
> **Reviewer 4 Response**
>
> Thank you for the review. We respond to your concerns below:
>
> - _“The authors only conduct experiments on small datasets, e.g., CIFAR. It would be stronger to report the comparison results on ImageNet.”_
> Please our general response, where we note that improving computer vision performance and large-scale evaluation on image recognition tasks is not a goal of this work.
>
> - _“The authors seek to allow users to design their own search spaces adapted to their data, which, however, is very labor-intensive. Moreover, it is also not clear how different search spaces would affect the search performance?”_
> We claim that a user can convert any fixed CNN backbone architecture, ideally a baseline model for their problem, into a NAS search space using K-operations. We disagree that this is labor-intensive because K-operations are a direct drop-in replacement for convolutions, so no further modifications to the backbone architecture are required. Furthermore, Supernet SGD is much more akin to standalone network training than standard NAS methods, and our search spaces don’t even necessarily require post-search retraining (offline evaluation). On the contrary, we argue that our search space design method and search methods are significantly less labor-intensive to the end user than current NAS. Furthermore, we provide code for users to easily replace convolutions with K-operations, and the API is nearly identical to the PyTorch nn.Conv2d API, which even further reduces the burden on the end user.
>
> - _“The proposed method fixes the backbone architecture and only searches for the optimal operation. However, this method ignores the topology of architectures. In my opinion, architecture topology should be an important factor to search in NAS.”_
> Our method intentionally fixes the backbone architecture to isolate the benefit of searching for operations. Our goal is not to find optimal architecture topologies, which can in principle be done in conjunction with K-operations anyway. We believe progress in NAS can be made in both directions separately, not always simultaneously.
>
> - _“The authors argue that the proposed method “enables the discovery of good design patterns with limited human specification from data in under-explored domains”. What does “under-explored domain” mean? How to define it?”_
> We are referring to domains which have not been studied much or at all in the NAS literature.
>
> - _“What is FFT in Page 4? There is no definition before it appears, which makes the paper hard to follow.”_
> Fast Fourier transform. We will disambiguate in the final version.
>
> - _“From Definition 3.2, the search process w.r.t. k-operations seems to learn the parameters of the matrix K1 and K2. Thus, it seems not a typical NAS process. It would be better to illustrate the relationship between the proposed method and the standard NAS.”_
> The architectural parameters are K1, K2, and M, not just K1 and K2. We illustrate the relationship between our proposed search space design and standard NAS in Section 2.
>
> - _“It is not clear what the proposed search space is. How large is it?”_
> The proposed search space is over the K-matrices K1, K2, and M, which are complex valued and continuous. Thus the search space has infinitely many architectures.
>
> - _“Several state-of-the-art NAS methods should be compared in the experiments, such as PDARTS, PCDARTS, etc.”_
> The evaluation of some NAS search methods, e.g. PC-DARTS, detracts from the point of our proposed NAS search space design method, or aren’t applicable to our search spaces, e.g. P-DARTS. The goal here is to design search spaces, not algorithms.
>
> - _“The writing of this paper should be improved. The paper is hard to follow and there are many grammatical errors: (1) In Page 3, “the main challenge … is intractability” should be “The main challenge … is intractable”. (2) In Page 5, “Dao et al. (2020) propose replace convolutional layers” should be “Dao et al. (2020) propose to replace convolutional layers””_
> Thank you for pointing this out. (1) both are grammatically correct, but we indeed mean to say that the main challenge is intractability. (2) has been fixed.

---

### Official Review · AnonReviewer1 · 2020-10-27
**An interesting method but with flaws**

**Rating:** 5
**Confidence:** 4

**Review:**

# Summary

The paper proposes to search neural network operations that outperform some human-designed ones, e.g. convolution. Specifically, it proposes to extend the Kaleidoscope paper to formulate a new searchable operation. Combining with differently architecture search method,

# Strength

The proposed Kaleidoscope Op is new to the field and seems to be an interesting aspect, searching for an operation similar to convolution but on different tasks.

# Major weakness

## 1. The presentation of this work hinders my understanding.

- The paper introduces a simple operation set but with many unnecessary descriptions and definitions. For example, what is the purpose to introduce definition 3.1? Does a parameterized operation return a parameterized function? This is confusing.

- Is Section 2 necessary? It occupies a significant space in the main text but is not related to the methodology section 3. I did not see any definition introduced in this section later. Instead, the definition of the core method of Kaleidoscope operations is not well introduced. The paper keeps mentioning the Dao 2020 and K-matrices, yet it only briefly defines it.

- What's the purpose to emphasize the three `key properties'? Will the author show some ablation study what happened if these properties are violated for some other operations? Why they are good properties in the NAS domain?

- How do you combine the NAS with K-Op (defined in 3.2)?
It is strange to see the authors spend a large space to describe the common NAS algorithms in section 2 but do not clearly state how this K-Op is combined with the search algorithms. On page 6, there is one sentence saying `K-op comprise a continuous search space...', but what exactly is that? What are the settings of your experiments? Please correct me if I miss something. For now, it should be clearly defined instead of a brief sentence to forester the reproducibility of this method.

- What's offline in Table 1? Does it mean you generate the architecture after search and train from scratch on CIFAR-10? But it seems on many methods the offline surpass the super-net, which is again weird.



## Weak baselines
This paper essentially extends the original paper to make it searchable. Does it seem to be reasonable to compare against it at least over one task in the Kaleidoscope paper? The baselines, e.g. LeNet, in this work are too weak compared to the recent NAS approach. I understand this paper does not claim to be state-of-the-art but showing the potential, yet with this baseline on CIFAR-10, it is hard to compare with the literature.


# Minor issues

- a random search on NAS on Page 7, did you mean Yu et al 2020 (evaluating the search phase of NAS) instead of Yang et al. 2020 (NAS Evaluation is frustratingly hard)? The latter describes the stand-alone training pipeline can significantly impact the reproducibility where Yu et al. describes that many NAS algorithms do not even outperform the random search.

- `FFT' on page 4 not defined when first used.
Broken reference on page 8.

- Repeating descriptions: Page 2 Top `desirable properties: short description length ...` with page 4 first paragraph of section 3.

---

> ### Author Response · Authors · 2020-11-20
> **Reviewer 1 Response (Part 2/2)**
>
> - _“This paper essentially extends the original paper to make it searchable. Does it seem to be reasonable to compare against it at least over one task in the Kaleidoscope paper?”_
> Our approach does not extend Dao et al. (2020) in making a searchable space out of the set of K-matrices, as they cannot express convolutional layers with multiple channels. On the other hand, K-operations do have channels across which architecture parameters are tied (i.e. the same operation is performed on each channel). Dao et al. (2020) instead focus on replacing specialized layers, e.g. permutations in ShuffleNet, rather than operations with channels. A version of permuted CIFAR-10 is presented in Dao et al. (2020), and we include results on a similarly permuted CIFAR-10 task in our work.
>
> - _“The baselines, e.g. LeNet, in this work are too weak compared to the recent NAS approach. I understand this paper does not claim to be state-of-the-art but showing the potential, yet with this baseline on CIFAR-10, it is hard to compare with the literature.”_
> We will include an evaluation using ResNet20 in the final version that shows that our approach also works using stronger backbone networks (see our general response for a results table).
>
> - _“a random search on NAS on Page 7, did you mean Yu et al 2020 (evaluating the search phase of NAS) instead of Yang et al. 2020 (NAS Evaluation is frustratingly hard)? The latter describes the stand-alone training pipeline can significantly impact the reproducibility where Yu et al. describes that many NAS algorithms do not even outperform the random search.”_
> Thank you. We will fix this citation.
>
> - _“`FFT' on page 4 not defined when first used. Broken reference on page 8.”_
> Thank you. We will fix this in the final version.
>
> - _“Repeating descriptions: Page 2 Top desirable properties: short description length ... with page 4 first paragraph of section 3.”_
> Thank you. We will fix this in the final version.

---

> ### Author Response · Authors · 2020-11-20
> **Reviewer 1 Response (Part 1/2)**
>
> Thank you for the review. We respond to your concerns below:
>
> - _“The paper introduces a simple operation set but with many unnecessary descriptions and definitions. For example, what is the purpose to introduce definition 3.1? Does a parameterized operation return a parameterized function? This is confusing.”_
> Parameterized operations indeed return parameterized functions, namely functions of the data parameterized by model weights. The purpose of definition 3.1 is to highlight the difference between parameterizable operations and simply learning a function of the data, as is done in Dao et al. (2020) and Neyshabur (2020). It is crucial to have a distinction between architecture parameters and model parameters - namely, the architecture parameters are the parameterizable operation itself while the model parameters are the parameters, w, of said operation.
>
> - _“Is Section 2 necessary? It occupies a significant space in the main text but is not related to the methodology section 3. I did not see any definition introduced in this section later. Instead, the definition of the core method of Kaleidoscope operations is not well introduced. The paper keeps mentioning the Dao (2020) and K-matrices, yet it only briefly defines it.”_
> We believe that Section 2 is indeed necessary for exposition of our search spaces because of the significant differences between our method of search space design and typical NAS over discrete search spaces. In particular, our search spaces are continuous, unlike virtually all of the current NAS literature and thus does not require post-search discretization. Furthermore, we find that performing offline evaluation is not always necessary, and that supernet training is sufficient, which is another significant difference between our method and current NAS methods (which we view as a benefit, as it is simpler). K-matrices require around 2 pages to define in Dao et al. (2020) and their specific structure is not critical for this work; what we require is their useful properties (efficiency, differentiability), which we do discuss. However, we will add an overview of K-matrices in the appendix of the final version.
>
> - _“What's the purpose to emphasize the three key properties'? Will the author show some ablation study what happened if these properties are violated for some other operations? Why they are good properties in the NAS domain?”_
> We emphasize the importance of these three properties to constrain search. Properties 1 and 3, short description length and fast computation, are chosen for efficiency because they reduce the effective size of the search space yet still yields a broad class of parameterizable operations to be recovered. Furthermore, these three properties are inherent to our approach, not hyperparameters that can be added or removed easily (how do you make a linear operation nonlinear? how can one construct an operation with O(n^2) description length without also getting O(n^2) time complexity?), so an ablation study does not make sense.
>
> - _“How do you combine the NAS with K-Op (defined in 3.2)? It is strange to see the authors spend a large space to describe the common NAS algorithms in section 2 but do not clearly state how this K-Op is combined with the search algorithms. On page 6, there is one sentence saying `K-op comprise a continuous search space...', but what exactly is that? What are the settings of your experiments? Please correct me if I miss something. For now, it should be clearly defined instead of a brief sentence to forester the reproducibility of this method.”_
> The search space is induced by K-Ops in Definition 3.2, where the architecture parameters are the 3 participant K-matrices. The search space is continuous because K-matrices occupy a continuous space of complex valued matrices. We point this out in Section 2 where we note that our search spaces do not require post-search discretization. To apply DARTS, for example, we use the bilevel objective (per Liu et al. 2018) and the same optimizers for the architecture and model parameters used in Liu et al. 2018. The only difference is that no post-search discretization is required. This is noted in Section 2 “The Goals of NAS.”
>
> - _“What's offline in Table 1? Does it mean you generate the architecture after search and train from scratch on CIFAR-10? But it seems on many methods the offline surpass the super-net, which is again weird.”_
> That is correct. Offline evaluation is described in Section 2 “The Goals of NAS.” It is not clear why offline evaluation sometimes improving upon supernet training is surprising, as this is the case for most NAS search spaces such as DARTS.

---

### Official Review · AnonReviewer3 · 2020-10-28
**Official Blind Review #3**

**Rating:** 5
**Confidence:** 4

**Review:**

Summary:

The paper introduces Kaleidoscope-operations to reprameterize convolutions. The reparameterization results in a more general search space of convolutions. Moreover, it enables one-level optimization on the convolution search as well as a supernet optimization, which avoids post-search discretization and post-discretization re-training.


Pros:

The paper seems a good reading material to teach readers about a big picture of convolution search problem or even neural architecture search problem.

The paper also shows a huge ambitious motivation to touch the boundary of the current main-stream NAS methodologies.

The idea of introducing repratermeterized convolutions (i.e., K-operations that was originally proposed by Dao et al., 2020) to convolution search seems novel and promising to me.

The evaluation is comprehensively conducted on several novel search spaces over vision and text data, and the results show the effectiveness of the proposed method. When being evaluated on permuted CIFAR and spherical MNIST, the new method shows some superiorities.


Cons:

It seems that the proposed model is designed to merely search for convolutions with a reprameterization approach (i.e, K-operation). Compared to regular NAS algorithms like DARTS that search for a much larger architecture space including convolutions, poolings, skip connections, this paper’s search space is merely on reprameterized convolutions. This makes me disappointed as both the title and the beginning parts somehow mislead readers that the paper aims at making a good innovation in the big scope of NAS. However, I finally realize that it actually searches for better convolutions rather than an entire neural architecture, after I went to the last paragraph of Page 5. Moreover, the search merely on convolutions is very likely to result in a much easier neural architecture optimization task, and make the so-called supernet (without post-search discretization & post-discretization retraining) work. This also reminds me that there exists one work [Stamoulis et al., 2019] which shares a similar motivation with this submission. In particular, [Stamoulis et al., 2019] proposes one single-path over-parameterized ConvNet to encode all architectural decisions with  convolutional kernel parameters. The overall network loss is directly a function of the “superkernel” weights, where the learnable kernel- and expansion ratio-related parameters can be directly derived as a function of the kernel weights. This strategy also enables one-level optimization and has a potential for supernet optimization as suggested by the submission.


[Stamoulis et al., 2019] Single-path NAS: Designing hardware-efficient convnets in less than 4 hours. In Joint European Conference on Machine Learning and Knowledge Discovery in Databases, 2019.


In Table 1, I find the best performances on CIFAR-10 and the transferring to CIFAR-100 are still far away from the state of the art. For example, DARTS (Liu et al., 2019) can obtain about 97% and 74% accuracies on CIFAR-10 and CIFAR-100 respectively. Please explain which causes this gap. For a fair comparison, I see the results of the proposed K-op with supernet SGD/SGDR are worse than Conv (fixed operation baselines, offline), while they are better when warm starting with convolution. What if warm-start is also applied to Conv (fixed operation baselines, offline)?

As presented in the paper, Fig.2 shows that learned K-operations use more global information to extract features. But what is the benefit for deep learning?


Table 2 is not self-contained. It should clarify the meaning of CR, MPQA, …, TREC in the caption. I guess they are the 7 used datasets according to the presentation in the main text. Again, I find the proposed method can obtain better results only when using the warm-start strategy, while it generally performs worse than the competitor (i.e., convolution) when training from scratch.

The last column of Table 4 seems confusing to me whether it corresponds to the case that uses warm start or from scratch.

In the paragraph of “utterfLeNet: Unpermuting Image Data”, the index of the referred section is missing.

Overall I think the paper oversells the new idea and the corresponding technology. Thus I tend to suggest a major revision by tuning down its current tone throughout the paper, while I like the idea and really expect the paper can be baked better for publication.

---

> ### Author Response · Authors · 2020-11-20
> **Reviewer 3 Response (Part 2/2)**
>
> - _“In Table 1, I find the best performances on CIFAR-10 and the transferring to CIFAR-100 are still far away from the state of the art. For example, DARTS (Liu et al., 2019) can obtain about 97% and 74% accuracies on CIFAR-10 and CIFAR-100 respectively. Please explain which causes this gap.”_
> The gap between our CIFAR-10 performance and that of DARTS is that we use a network with 2 conv layers as a backbone, whereas at evaluation DARTS uses a network that can have 20 or more conv layers. We will add an evaluation with a ResNet20 backbone (9 conv layers) in the final version showing that our approach also works with deeper networks (see our general response for a results table); however, as pointed out in our general response achieving SOTA on computer vision tasks is not a goal of this paper.
>
> - _“For a fair comparison, I see the results of the proposed K-op with supernet SGD/SGDR are worse than Conv (fixed operation baselines, offline), while they are better when warm starting with convolution. What if warm-start is also applied to Conv (fixed operation baselines, offline)?”_
> Warm-start is not applicable to Conv because warm-start refers to warm-starting the architecture parameters as convolutions. Conv does not have any architecture parameters because it is a fixed operation. In both from-scratch and warm-start settings the actual model weights are initialized randomly.
>
> - _“As presented in the paper, Fig.2 shows that learned K-operations use more global information to extract features. But what is the benefit for deep learning?”_
> The point of this figure is to show that we can search the space to find operations very distinct from convolutions. The benefit of this is that in non-vision data domains we may learn very different types of feature extractors.
>
> - _“Table 2 is not self-contained. It should clarify the meaning of CR, MPQA, …, TREC in the caption. I guess they are the 7 used datasets according to the presentation in the main text. Again, I find the proposed method can obtain better results only when using the warm-start strategy, while it generally performs worse than the competitor (i.e., convolution) when training from scratch.”_
> Yes, these are names of datasets in a standard NLP evaluation; we will clarify this in the final version. As we noted above, warm-starting architecture parameters is not applicable for the Conv setting, which does not have architecture parameters.
>
> - _“The last column of Table 4 seems confusing to me whether it corresponds to the case that uses warm start or from scratch.”_
> Here the first K-op layer is searched from-scratch while the second is warm-started with convolutions. This is described in the Section 5 paragraph titled “ButterfLeNet-Spherical” and we will also add a clarification to the caption in the final version.
>
> - _“In the paragraph of “utterfLeNet: Unpermuting Image Data”, the index of the referred section is missing.”_
> Thank-you, we will fix this in the final version.
>
> - _“Overall I think the paper oversells the new idea and the corresponding technology. “_
> We disagree that the paper is overselling; there is instead a misunderstanding of the goals and contributions. Our search space contains many more operations than just convolutions, and our goal is not to improve SOTA in computer vision but to push NAS to discover the “right” operation to use in new application domains.

---

> ### Author Response · Authors · 2020-11-20
> **Reviewer 3 Response (Part 1/2)**
>
> Thank you for your detailed review. We respond to your concerns below:
>
> - _“It seems that the proposed model is designed to merely search for convolutions with a reparameterization approach (i.e, K-operation). Compared to regular NAS algorithms like DARTS that search for a much larger architecture space including convolutions, poolings, skip connections, this paper’s search space is merely on reparameterized convolutions. This makes me disappointed as both the title and the beginning parts somehow mislead readers that the paper aims at making a good innovation in the big scope of NAS. However, I finally realize that it actually searches for better convolutions rather than an entire neural architecture, after I went to the last paragraph of Page 5. Moreover, the search merely on convolutions is very likely to result in a much easier neural architecture optimization task, and make the so-called supernet (without post-search discretization & post-discretization retraining) work.”_
> The space of operations is *not* a reparameterization of convolutions; for example, search over the space of convolutions would not be able to solve the permuted image task because such as space does not contain permutations, whereas our space contains architectures that are able to do well. In fact, our architecture search space is actually much larger than ones such as DARTS, as ours contains infinitely many possible operations, and more difficult, in the sense that randomly selected architectures do poorly. We disagree that the first part of the paper was misleading: in the abstract we directly say that we want to “discover the right neural operations given data” which restricts the scope to operations, not entire architectures. We believe NAS breakthroughs do not have to come from immediate application to state-of-the-art computer vision by searching entire architectures, but can also come from search space designs that push the field beyond this domain to settings where convolutions do not work well; the latter is the goal of our work. As an example, AutoML-Zero (Real et al., ICML 2020) is another example of this.
>
> - _“This also reminds me that there exists one work [Stamoulis et al., 2019] which shares a similar motivation with this submission. In particular, [Stamoulis et al., 2019] proposes one single-path over-parameterized ConvNet to encode all architectural decisions with convolutional kernel parameters. The overall network loss is directly a function of the “superkernel” weights, where the learnable kernel- and expansion ratio-related parameters can be directly derived as a function of the kernel weights. This strategy also enables one-level optimization and has a potential for supernet optimization as suggested by the submission.”_
> The goal of Stamoulis et al. (2019) is to find hardware-efficient convolutional networks that do well on computer vision data, whereas our goal is to search a space of operations much larger than convolutions to do well on both vision and non-vision data.

---

### Official Review · AnonReviewer2 · 2020-10-30
**An interesting paper**

**Rating:** 5
**Confidence:** 5

**Review:**

The paper claims to perform neural operator search on a search space defined by a family of Kaleidoscope operations. The paper address the computation challenges in the search using  a supernet, and performs variable ablation studies to show that the searched K-Op can slightly outperforms existing convolution operators.

Here I appreciate if the authors can clarify the following points:

a) "Each butterfly matrix of dimension n x n is itself a product of log n sparse matrices with a special fixed sparsity pattern, which encodes the recursive divide-and-conquer algorithms such as the FFT."

Here the author deliberately selects a family of operations that contains FFT and convolution. It is unclear how you constrain the search space of K-matrices; I get you searched K operations, but I did not find it's structure and how does that different from FFT. I'm also confused about the point of fig.2, what's the point you want to say about these feature map?

b) Ablation studies.

There are a lot of ablation studies show that the searched K-operations are better than convolution. However, none of them show they actually pushed SoTA results. Considering the fact that there are so many tricks (https://github.com/facebookresearch/LaMCTS/tree/master/LaNAS/LaNet) in boosting the performance of a CNN, it is more convincing to see the searched K operators can actually push the boundary. The ablation studies in the current experiments are not enough to convince me, especially they are focusing on relatively simple tasks, e.g. CIFAR-10, MNIST.

c) Another thought, Tensorized Neural Network has also tried to replace of current operators, and their hyper-parameters can also formulate a search space; My main concern about this line of work is "are we really making progress here"?

Here we're building something based one prior knowledge; if we will end up someting similar to convolution, so what's the point of doing it? However, the paper lacks a strong evidence that they invented a new operators that actually work. This is my main concern of this paper.

---

> ### Author Response · Authors · 2020-11-20
> **Reviewer 2 Response**
>
> Thank you for your review. We hope to address your questions below:
>
> - _“"Each butterfly matrix of dimension n x n is itself a product of log n sparse matrices with a special fixed sparsity pattern, which encodes the recursive divide-and-conquer algorithms such as the FFT." Here the author deliberately selects a family of operations that contains FFT and convolution. It is unclear how you constrain the search space of K-matrices; I get you searched K operations, but I did not find it's structure and how does that different from FFT.”_
> A K-matrix is a generalization of FFTs and is defined as a product of matrices with a fixed sparsity structure, which directly constrains the search space; unlike FFTs, K-matrices can express any matrix whose matrix-vector product takes linear time to express/compute. A detailed construction is given in Dao et al. (2020), who introduced them; the contribution of this paper is to use them to define an architecture search space containing convolutions and other fast operations. The search space consists of layers defined as $f_w(x)=Real(K\_1\*diag(M\*w)\*K\_2\*x)$, where $K\_1$, $K\_2$, and $M$ are K-matrices. In the case where $f\_w$ is a convolution with filter weights w these K-matrices are defined using FFTs, but that is just one possible architecture in the search space over $K\_1$, $K\_2$, and $M$.
>
> - _“I'm also confused about the point of fig.2, what's the point you want to say about these feature map?”_
> Figure 2 illustrates both that K-operations can be found that behave very similarly to convolutions (center plot) or very differently (right plot) given the same input, demonstrating their flexibility as feature extractors. The point is to illustrate what kind of operations can be found.
>
> - _“There are a lot of ablation studies show that the searched K-operations are better than convolution. However, none of them show they actually pushed SoTA results. Considering the fact that there are so many tricks (https://github.com/facebookresearch/LaMCTS/tree/master/LaNAS/LaNet) in boosting the performance of a CNN, it is more convincing to see the searched K operators can actually push the boundary. The ablation studies in the current experiments are not enough to convince me, especially they are focusing on relatively simple tasks, e.g. CIFAR-10, MNIST.”_
> Please see Point 1 of our general response, where we note that improving computer vision performance is not a goal of this work.
>
> - _“Another thought, Tensorized Neural Network has also tried to replace of current operators, and their hyper-parameters can also formulate a search space; My main concern about this line of work is "are we really making progress here"?”_
> It is unclear what approach “Tensorized Neural Networks” refers to. A top search result is a paper (https://arxiv.org/abs/1509.06569) in which replacing convolutions by tensors outperforms only non-convolutional networks, not convolutions; in contrast, we can learn operations that are better than convolutions.
>
> - _“Here we're building something based one prior knowledge; if we will end up something similar to convolution, so what's the point of doing it? However, the paper lacks a strong evidence that they invented a new operators that actually work. This is my main concern of this paper.”_
> We respectfully disagree with this statement. Our paper demonstrates on permuted images and spherically projected images that it can discover operations that work much better than convolutions on those domains. It is not surprising that our operations are only slightly better and similar to convolutions on computer vision data, since convolutions are known to work well there; however, as pointed out in our general response the goal of our paper is to push NAS beyond computer vision tasks.

---

### Author Response · Authors · 2020-11-20
**General response**

Thank you to the reviewers for providing feedback. We respond to individual comments separately, but first would like to discuss  a common concern among several reviewers, specifically the question of why the paper does not improve SOTA in computer vision/evaluate on large-scale dataset such as ImageNet. We emphasize that the goal of this paper is **not** to improve performance on computer vision benchmarks; in contrast, as pointed out in the abstract our goal is to push NAS research beyond computer vision tasks and simple search spaces to new application domains where convolutions might not be the “right” feature extractor to use. To this end, the point of our experiments on simple computer vision tasks is to show on an established benchmark that our techniques are competitive with and can even beat convolutions, which is evidence that they can discover the “right” feature extractors in new domains (which we also experiment with explicitly on NLP data, permuted images, and spherical images).

Furthermore, in the final version, we will upload new results showing that our method also improves the performance of more competitive networks such as ResNets on CIFAR data. We summarize the results of our ResNet experiments on CIFAR10, averaged over 5 trials, in the following table:



|          | Conv   | Supernet | Offline |
|----------|--------|----------|---------|
| ResNet8  | 87.378 | 89.85    | 89.47   |
| ResNet14 | 90.28  | 91.76    | 91.71   |
| ResNet20 | 91.34  | 92.58    | 92.33   |


For each of these, we warm start all of the K-operations as convolutions and fix the operations to be convolutions for the first 50 out of 200 total epochs, which amounts to setting the architecture learning rate to 0 for the first 50 epochs. We found that this change to the architecture learning rate schedule was not always necessary to outperform convolutions in supernet evaluation, but it further improves performance nonetheless. Note that the parameter count for the columns Conv and Offline are the same, hence the performance gain is indeed due to better operations found by Supernet SGD.

---

### Decision · Program_Chairs · 2021-01-07
**Final Decision**

**Decision:**

Reject

**Comment:**

Motivated by the possibility of Neural Architecture Search on domains beyond computer vision, this paper introduces a new search space and search method to improve neural operators. It applies the technique to problems in vision and text.

Reviewer 1 found the paper interesting and liked the motivation of considering different tasks in NAS. However, they found some aspects of the paper confusing and, like other reviewers, thought that the baselines were weak. The authors clarified some points, and R1 said that some, but not all, concerns were resolved. The reviewer improved their score by a point but still was not in favour of acceptance.

Reviewer 2 thought the paper was interesting but questioned its main claim: that it was proposing a search space over novel operators. They argued that what was discovered was similar to convolution and therefore not much had been gained. They questioned the significance of the ablation studies: there were a lot of them, but they focused on relatively simple tasks like MNIST and CIFAR-10. They also asked some clarifying questions which were answered by the authors. Pushing back on the point about the smaller scale of the experiments (in a general reply to all reviews), the authors said that the goal of their work was not advancing computer vision, but to push NAS beyond computer vision and simple search spaces to new application domains.

Reviewer 3 liked that the paper gave a good overview of the NAS problem and thought that it was ambitious. They also thought the approach was novel and promising. Like R2’s comment, R3 seemed disappointed that the search was over “reparameterized convolutions”. In fact, they thought that the paper was overselling its contribution. They pointed out that performance was still far from state-of-the art on the various benchmarks. The authors argued against this view of “reparameterizing convolutions” and claimed that the search space was, in fact, much larger than that of DARTS. R3 read and responded to the rebuttal, appreciating the response but ultimately thought that the search space wasn’t clear and comparisons fell short.
Reviewer 4 shared similar pros & cons as have been pointed out by the other reviewers. They thought that operator search was limiting and that the paper should also consider topology. The authors responded to this, saying that they intentionally fixed the topology. Searching beyond operators was out of scope. R4 responded to the rebuttal though still had some remaining concerns both in terms of motivation and execution.

Multiple reviewers said that they would have considered the paper more favourably had an updated paper been submitted, addressing some of the original concerns. As it stands, all of the reviewers think that the paper has some merits but none believe after considering the author response, that the paper is ready for acceptance. I see no reason to overrule the consensus.